# Melting and defect transitions in FeO up to pressures of Earth's core-mantle boundary

Vasilije V. Dobrosavljevic [1,2] ✉, Dongzhou Zhang [3], Wolfgang Sturhahn[1], Stella Chariton [4], Vitali B. Prakapenka [4], Jiyong Zhao[5], Thomas S. Toellner[5], Olivia S. Pardo[1,6] & Jennifer M. Jackson[1]

The high-pressure melting curve of FeO controls key aspects of Earth's deep interior and the evolution of rocky planets more broadly. However, existing melting studies on wüstite were conducted across a limited pressure range and exhibit substantial disagreement. Here we use an in-situ dual-technique approach that combines a suite of >1000 x-ray diffraction and synchrotron Mössbauer measurements to report the melting curve for $Fe_{1-x}O$ wüstite to pressures of Earth's lowermost mantle. We further observe features in the data suggesting an order-disorder transition in the iron defect structure several hundred kelvin below melting. This solid-solid transition, suggested by decades of ambient pressure research, is detected across the full pressure range of the study (30 to 140 GPa). At 136 GPa, our results constrain a relatively high melting temperature of $4140 \pm 110$ K, which falls above recent temperature estimates for Earth's present-day core-mantle boundary and supports the viability of solid FeO-rich structures at the roots of mantle plumes. The coincidence of the defect order-disorder transition with pressure-temperature conditions of Earth's mantle base raises broad questions about its possible influence on key physical properties of the region, including rheology and conductivity.

Wüstite ($Fe_{1-x}O$) has long been recognized for its prominent role in controlling the properties and evolution of Earth and other rocky planetary bodies[1–4]. In particular, FeO represents an end-member component in Earth's major mineralogical systems, with its melting point being an essential parameter for constructing planetary interior models. In the $FeO-MgO-SiO_2$ system of the mantle, the melting curve of FeO controls crystallization sequences of Earth's primordial magma ocean[3,5–9]. The Fe-FeO system has been extensively studied to assess the viability of oxygen as a major light element in Earth's outer core[5,10–12]. FeO has further been implicated in chemical and heat exchanges between the core and mantle[13,14], as well as in the deep mantle water cycle[15] over geologic time.

In the last decade, the properties of FeO have received renewed attention in the context of ultralow velocity zones, enigmatic regions of extremely reduced seismic wave speeds dispersed across Earth's heterogeneous mantle base[16], co-located at edges of large thermochemical piles and at roots of major mantle plumes that source volcanic hotspots like Hawai'i, Iceland, and Yellowstone (e.g., refs. [17–20]). Recent work has suggested that these structures, originally posited to consist of partial melt[21], can be explained by the presence of solid (Mg,Fe)O with high concentrations of FeO[22–27], leading to remarkably low seismic velocities[28,29], low viscosity[30], high seismic anisotropy[31], and high conductivity[32].

[1]Seismological Laboratory, Division of Geological and Planetary Sciences, California Institute of Technology, Pasadena, CA, USA. [2]Now at Earth and Planets Laboratory, Carnegie Institution for Science, Washington, DC, USA. [3]Hawai'i Institute of Geophysics and Planetology, University of Hawai'i at Mānoa, Honolulu, HI, USA. [4]Center for Advanced Radiation Sources, The University of Chicago, Chicago, IL, USA. [5]Advanced Photon Source, Argonne National Laboratory, Chicago, IL, USA. [6]Now at Physics Division, Physical & Life Sciences Directorate, Livermore, CA, USA. ✉e-mail: vasilije@carnegiescience.edu

The melting curve of FeO, however, has remained highly uncertain, especially at pressures of the deep lower mantle and core, presenting a major obstacle to understanding these various geophysical and geochemical systems. Most experimental measurements at high pressures have relied on proxy phenomena, like changes in sample emissivity or quenched sample textures, and exhibit substantial disagreement (more than 700 K at 70 GPa)[6,7]. Recent extrapolations to lowermost mantle pressure (136 GPa) from experimental work and thermodynamic calculations exhibit a similarly extreme uncertainty range[5] that partially overlaps with suggested core-mantle boundary temperatures (e.g., ref. 33). This makes it impossible to determine the viability of solid FeO-rich structures in the region and introduces large uncertainties into models of Earth's thermochemical evolution.

Investigation of the phase diagram is further complicated by the presence of iron defects, $Fe_{1-x}O$. Studies at ambient pressure have reported the formation of short-range defect clusters, consisting of $Fe^{2+}$ vacancies and interstitial $Fe^{3+}$ atoms, which can develop into long-range periodic superstructures within the $Fe_{1-x}O$ lattice at moderately elevated temperatures[2,34–38]. At higher temperatures, but below melting, some studies suggest a defect order-disorder transition could occur in $Fe_{1-x}O$[2,39], as well as in related materials like $Fe_{1-x}S$[40,41] and $Fe_{1-x}Se$[42] with implications for superconductivity[43,44]. However, no studies have explored such iron defect transitions at simultaneous high pressures and temperatures nor investigated possible consequences for precise determination of melting temperatures.

In this study, we investigate the behavior of $Fe_{0.94}O$ at simultaneous high pressures and temperatures using a recently developed multi-technique approach[45] that combines results from two in-situ techniques that probe different length and time scales—synchrotron x-ray diffraction (XRD), sensitive to atomic positions, and synchrotron Mössbauer spectroscopy (SMS), sensitive to dynamics of the iron atoms. We systematically survey the phase diagram of $Fe_{0.94}O$ from 30 to 140 GPa and 300 to 4500 K with a suite of ~1000 x-ray diffraction images and ~200 synchrotron Mössbauer measurements collected over 23 separate heating runs at beamlines 13-ID-D and 3-ID-B of the Advanced Photon Source, respectively. Details of experimental procedures can be found in Methods.

## Results and discussion
### Observations in x-ray diffraction patterns

For all heating runs across the entire pressure range, every reflection observed at the start of heating can be indexed by $B1$-$Fe_{1-x}O$ at high temperatures (>1200 K, up to 140 GPa), $rB1$-$Fe_{1-x}O$ at 300 K (up to 128 GPa), and $B2$-KCl at all temperatures. As the heating temperature increases during each run, we observe the emergence of additional reflections for both high temperature ("hot") measurements and quenched measurements taken iteratively after each hot one (Fig. 1, S1). These reflections, seen in both the integrated patterns and the raw XRD images, are located at similar azimuthal angles as $B1$ and $rB1$-$Fe_{1-x}O$ reflections with small offsets in the $2\theta$ scattering angle. Often referred to as satellite reflections, these types of reflections have been commonly observed in previous studies on $Fe_{1-x}O$ and attributed to long-range ordering of iron defects at ambient[2,38] and high pressure[46]. In particular, we consistently observe satellites around the $(2\,0\,0)$ $B1$-FeO reflection, as seen in previous studies[46]. The $d$-spacings of these satellites relative to the $(2\,0\,0)$ reflection appear essentially constant across the full pressure range of the study (Fig. 2, Methods), suggesting they are produced by features of the defect-bearing $Fe_{0.94}O$ lattice and not a separate phase.

We identify several key trends that systematically appear across the ensemble of XRD heating runs (Fig. 3, S2–6). Firstly, we see strong anti-correlation in the intensities of the $Fe_{1-x}O$ satellite reflections and the background signal, for both hot and quenched measurements. Specifically, with increasing temperature in the hot measurements, we observe the onset and increase in intensity of satellite reflections,

before a reduction in intensity at a temperature $T_1$ (Fig. 3). Simultaneously, the hot background intensity gradually decreases from its starting value before exhibiting a small increase or plateau in intensity at the same temperature $T_1$. We constrain $T_1$ using both of these observations, with uncertainties estimated from scatter in the data. In the quenched measurements, we observe a similar increase in satellite reflection intensity that plateaus at relatively high intensities at $T_1$, while the background intensity gradually decreases from its starting value and reaches a minimum at $T_1$. In many heating runs, sample temperatures additionally exhibit plateau-like behavior in their dependence on laser power as the sample reaches $T_1$.

In more than half of the heating runs, the sample is heated above the $T_1$ temperature plateau and reaches a second transition temperature $T_2$. Here, hot measurements continue to show no satellite intensities, whereas the hot background signal now begins increasing steeply to much larger values, suggesting the onset of a diffuse signal from liquid melt (Fig. 3, S1). We constrain $T_2$ using this diffuse signal onset, with uncertainties from scatter in the data. In contrast, quenched measurements show some loss of satellite intensity and an increase of the quenched background signal intensity back to its starting value when quenching from $T_2$.

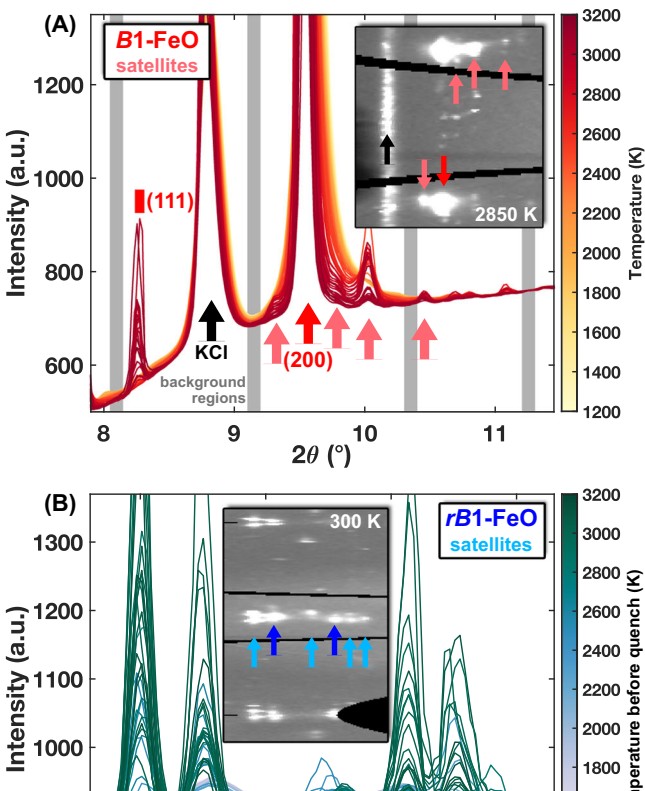

**Fig. 1 | Integrated diffraction patterns from a typical XRD heating run.** Run 21D1S1 at (**A**) high temperatures and (**B**) quenched to 300 K. Colored arrows indicate $Fe_{1-x}O$ sample (darker) and satellite (lighter) reflections. Insets show portions of azimuthally unwrapped (caked) 2D diffraction images from representative measurements in the heating run. Inset images plot scattering intensity as a function of scattering angle $2\theta$ (horizontal) and azimuthal angle $\alpha$ (vertical), showing the same satellite and sample reflections indicated by arrows in the integrated patterns. Gray bars indicate integration regions for background intensity analysis. Positions of $(2 \pm \delta\,0\,0)$ satellite reflections (pink arrows) are analyzed in Fig. 2 for all heating runs.

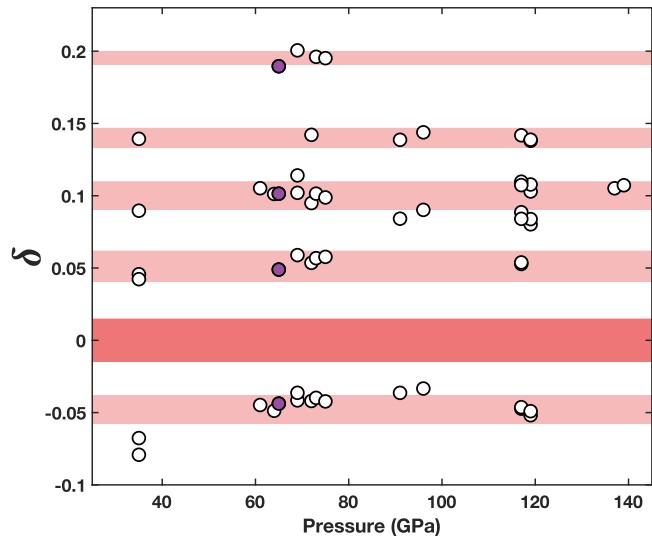

**Fig. 2 | Positions of $(2 \pm \delta\ 0\ 0)$ satellite reflections (circles) from all XRD heating runs.** Values of $\delta$ are calculated from the $d$-spacing of each satellite and the lattice parameter $a$ for each heating run using the formula $\frac{1}{d^2} = \frac{h^2 + k^2 + l^2}{a^2}$, corresponding to the cubic lattice system. Purple circles identify satellites marked by pink arrows in Fig. 1 (heating run 21D1S1). Shaded bands show the typical full-width half maxima of satellite (pink) and $B1$-(2 0 0) (red) reflections. Note the constant relative positions of satellites across the full range of pressures (sample volumes).

The intensities of the primary $Fe_{1-x}O$ reflections also show distinct behavior at $T_1$ and $T_2$. $B1$ (hot) and $rB1$ (quenched) intensities exhibit fluctuations associated with the onset of satellite reflections and scattering around the starting intensity at $T_1$. Above $T_2$, $B1$ intensities drop to negligible values (~10–20% of starting intensity), while $rB1$ intensities remain large. Quenched samples in all heating runs exhibit the $rB1$ structure except in the two highest pressure runs, where quenching from $T_2$ preserves the $B1$ structure at ~125 GPa. We do not observe evidence of the $B8$ structure within detection limits at any of the $P$–$T$ conditions explored in this study.

## Interpretation of x-ray diffraction observations

We interpret the observed trends as consequences of iron defects arranged in long-range ordered superstructures at moderate temperature, a defect structure order-disorder transition in the solid sample at $T_1$, and melting of the $B1$-$Fe_{1-x}O$ lattice at $T_2$. Specifically, we suggest that the sample initially features disordered defects (iron vacancies that may be compensated by ferric $Fe^{3+}$ in tetrahedral interstitial or octahedral cation sites) which then progressively develop long-range ordering with increasing temperatures, leading to an anti-correlated increase in satellite intensity and small decrease in background signal in both hot and quenched measurements. At temperatures above $T_1$, the iron defects lose their long-range order and transition to a disordered state, leading to a loss of satellite intensities and small increase in background intensity. Importantly, the sample

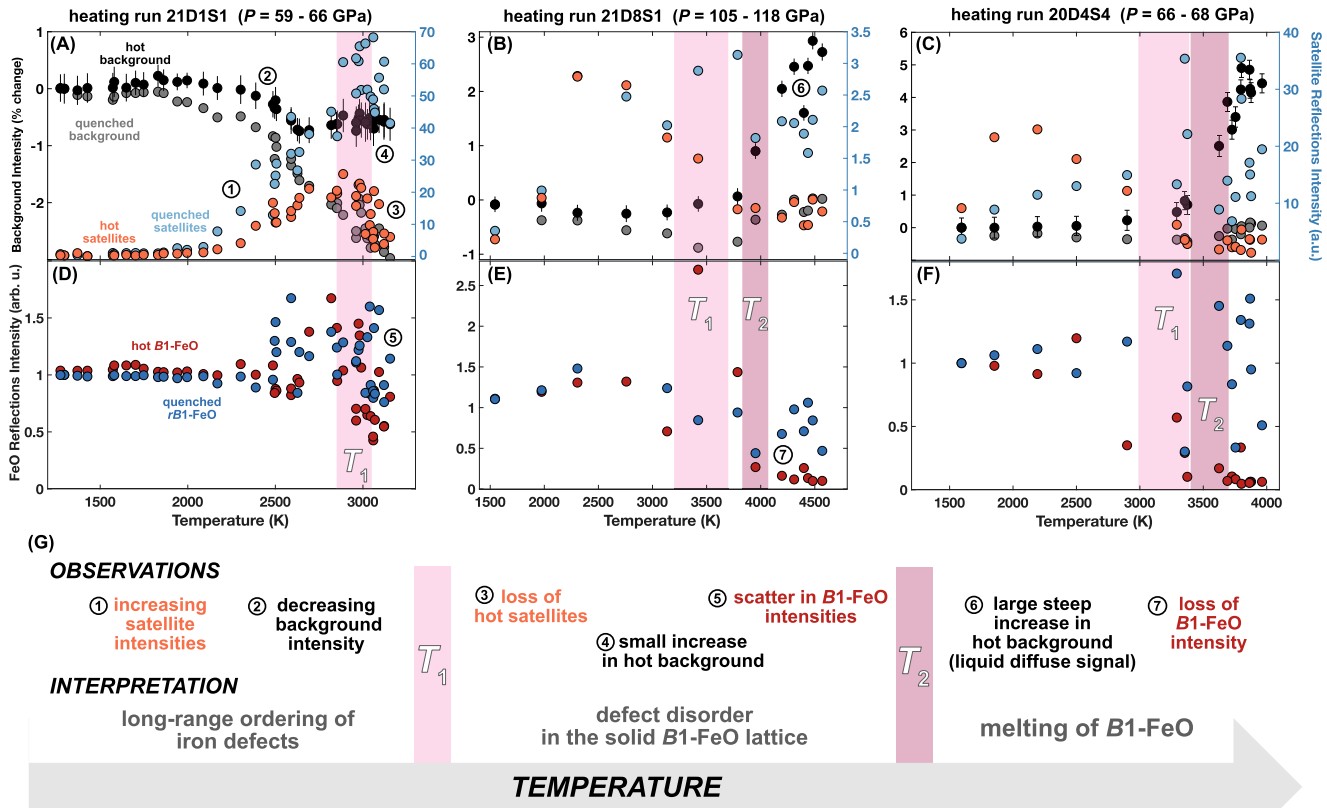

**Fig. 3 | Observed trends in XRD heating runs and interpretation.** Three representative XRD heating runs (**A**–**F**) showing temperature dependence of the intensities of background signals, satellite reflections, and $Fe_{1-x}O$ reflections. Shaded bars indicate the defect order-disorder (light, $T_1$) and melting (dark, $T_2$) transitions. The schematic (**G**) highlights key observations for each portion of the heating runs with resulting interpretation.

remains solid at $T_1$, evidenced by the persistence of $B1$-$Fe_{1-x}O$ reflections and small changes (<1%) in background signal.

We thus interpret these intensity variations of the background around the low-order Bragg reflections (Fig. S1, Methods) as consequences of defects, whose disorder in the solid sample may contribute a small diffuse signal to backgrounds in both hot and quenched patterns. Only above $T_2$ do we see clear evidence for melting—significant (>>1%) and steep increase in background intensity, interpreted as liquid diffuse scattering, and a loss of intensity from the sample's Bragg reflections. We note that the background intensities show no correlation with sample's primary reflection intensities except above $T_2$, where we interpret the correlated large hot background signals and loss of sample reflections as evidence of melting.

The observed trends and suggested interpretation in this study mirror similar observations and interpretations for the high-temperature behavior of $Fe_{1-x}O$ at ambient pressure. Satellite reflections in $Fe_{1-x}O$ have been extensively studied at ambient pressure and consistently attributed to long-range ordered defect structures, while such relatively small levels (compared with that of a liquid signal) of diffuse scattering has been commonly understood as long-range disorder and possible short-range clustering of defects[2,38]. In particular, the development of long-range defect ordering at moderate temperatures and the transformation to a disordered state at high temperatures several hundred kelvin below melting have been suggested for $Fe_{1-x}O$ at ambient pressure[2] but never studied at high pressures. The findings in this study provide evidence for similar behavior at simultaneous high pressures and temperatures.

## Determination of melting and ferric iron content with synchrotron Mössbauer spectroscopy

Melting temperatures of $Fe_{0.94}O$ are independently determined using time-domain synchrotron Mössbauer spectroscopy (SMS). The Mössbauer signal, produced exclusively by the nuclear resonance of solid-bound $^{57}Fe$ atoms with negligible background, manifests as photons scattered in the forward direction that are delayed (due to the nuclear excitation lifetime) relative to the much more intense non-resonant scattering from the sample's electrons[33,47]. The intensity of time-delayed (>20 ns) photons is recorded in 3-s intervals as the sample is heated with a ramped laser power sequence, leading to the loss of signal intensity at the onset of melting (Fig. 4 and S7, 8). The melting temperature is quantitatively constrained by fitting the temperature dependence of the Mössbauer signal (time-integrated delayed counts) with an experiment-specific scattering intensity model that incorporates measurements of the x-ray beam and laser hotspot sizes (see Methods). Melting temperatures determined in the SMS heating runs exhibit good agreement with the $T_2$ temperatures constrained in the XRD measurements within mutual uncertainties (Tables S2, 3). This finding provides clear corroboration of the interpretation of melting at $T_2$ in the XRD measurements, especially given strong agreement in melting temperatures of $Fe_{0.8}Ni_{0.1}Si_{0.1}$ previously determined by these two techniques at the same beamlines as in this study[45]. We further report a heating run that reached a plateau temperature without a loss of Mössbauer signal intensity (Fig. S9), suggesting the sample remained solid at the maximum temperature of the heating run. This temperature aligns closely with the $T_1$ temperatures constrained in the XRD measurements (see below), providing further corroboration of $T_1$ as a solid-solid transition and not the onset of melting.

We additionally measure Mössbauer time spectra that provide information on sample thickness and the electronic environment of the $^{57}Fe$ atoms (Fig. 5). We collect a spectrum before each SMS melting run while annealing the sample at ~1500 K, as well as after one SMS melting run. We further measure spectra for the starting sample material at ambient conditions as well as for several samples from completed XRD heating runs that we decompress to low pressures (<5 GPa) where FeO does not exhibit magnetic ordering. We fit this

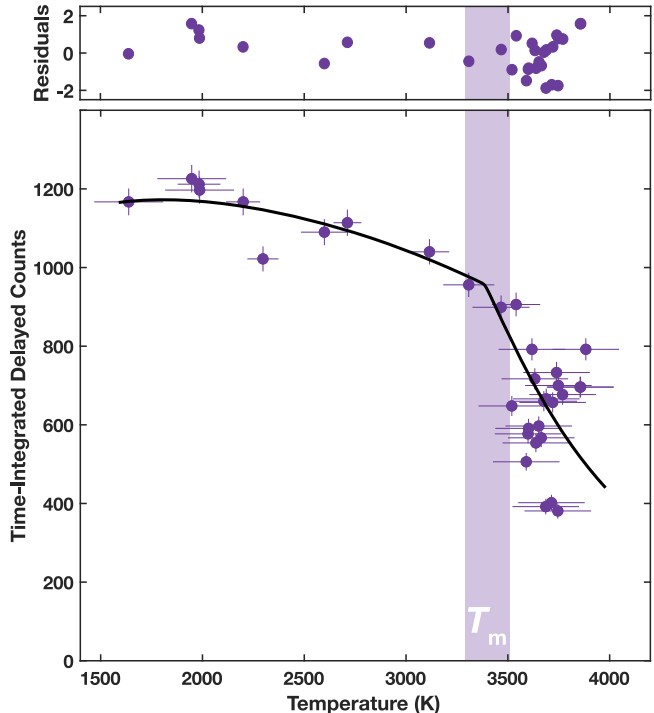

**Fig. 4 | Typical SMS heating experiment showing Mössbauer signal intensity (time-integrated delayed counts) of $Fe_{1-x}O$ as a function of temperature.** Fit to the data (solid black line) with an experiment-specific scattering intensity model constrains the melting temperature (shaded bar) for heating run 18D6S1 ($P = 50$–$59$ GPa). Residuals are plotted in units of standard deviation.

suite of time spectra using a consistent electronic model (see Methods) that provides information on how ferric iron content (and thus implied defect concentration) changes during heating and melting.

We report several key findings from fits to the Mössbauer time spectra (Fig. 5 and S10–13, Tables S4–14). Firstly, we constrain a ferric ($Fe^{3+}$) iron concentration of $11.2 \pm 0.4\%$ in the starting material, generally consistent with the $Fe_{0.94}O$ composition estimated from the ambient-pressure unit-cell volume and with unheated or annealed sample regions. Secondly, we observe an increase in the $Fe^{3+}$-like site after melting, seen both in high-$P$-$T$ spectra after an SMS melting run (Fig. 5A) and in melted regions from XRD heating runs in decompressed samples (Fig. 5B), suggesting preferential partitioning of ferric iron into the melt, or a change in defect structure affecting the ensemble electronic environment.

Finally, the ferric-like line is broad (a full-width at half-maximum, FWHM ~ 1 mm/s) in the starting material and unheated sample regions but narrower (FWHM ~ 0.4 mm/s) in melted regions and narrowest (FWHM ~ 0.2 mm/s) in regions heated below melting. This suggests $Fe^{3+}$ atoms are initially distributed across a wide range of electronic environments, which then become more homogeneous with heating. This parallels the findings from XRD measurements showing that an initially disordered defect structure becomes progressively ordered during heating, with the ordered structures preserved in quenched samples producing satellite reflections in XRD data and narrow $Fe^{3+}$ lines in SMS spectra.

## Phase diagram of $Fe_{0.94}O$

We present a summary of all results from this study on melting and order-disorder transition temperatures for $Fe_{0.94}O$ in Fig. 6 and Tables S1–3. Transition pressures are determined in-situ for XRD heating runs using primary lattice cell volumes and previously published equations of state[48–50] and are calculated for SMS runs from a thermal pressure model constructed from select XRD runs in this study

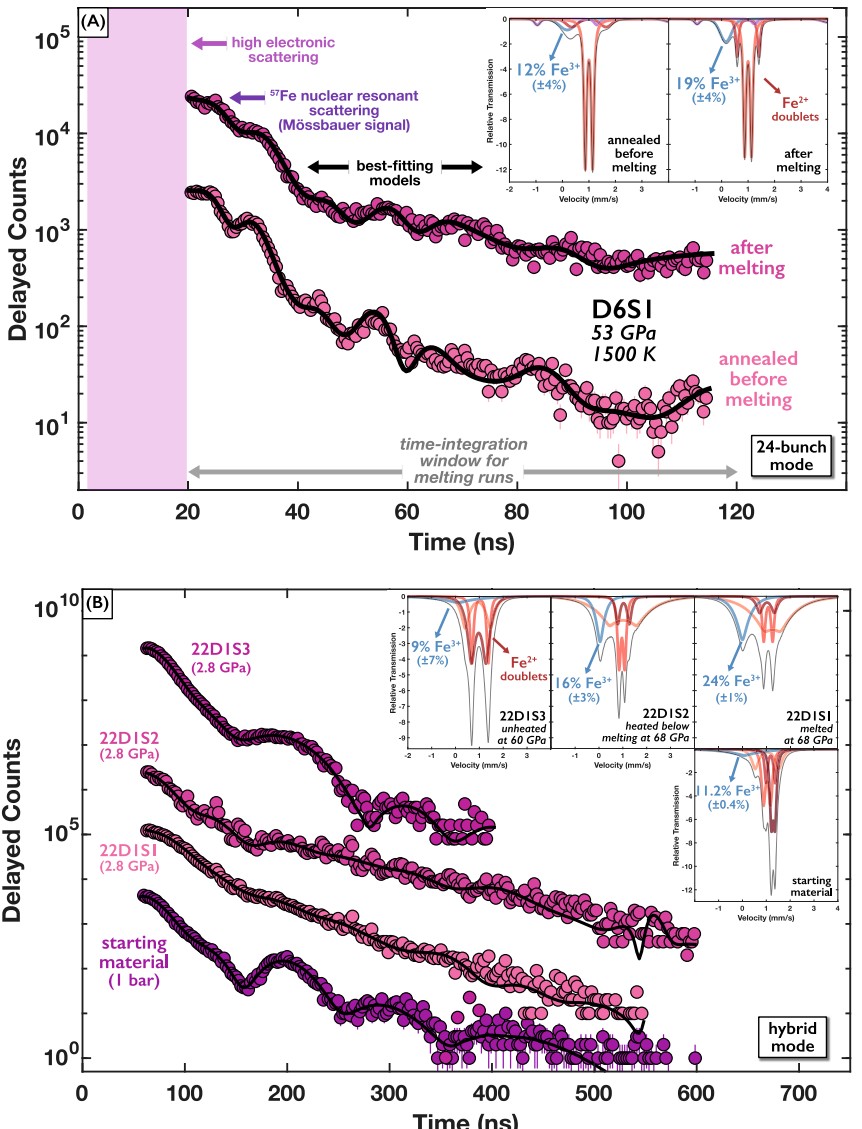

**Fig. 5 | SMS time spectra showing the effect of heating and melting on the electronic environments of $^{57}$Fe in Fe$_{1-x}$O. A** 24-bunch top-up mode spectra before and after SMS melting run 18D6S1 (see Fig. 4) showing data (colored points) and best-fitting models (black lines). **B** Hybrid mode spectra for the starting material and a sample (22D1) heated at Sector 13 (XRD) and decompressed to low pressure. Insets show corresponding best-fitting models calculated in the energy domain, highlighting ferric iron lines (blue) with labeled relative weights. Insets in (**A**) feature a truncated horizontal axis for clarity, with the magnetically ordered site shown in purple. In (**A**) and (**B**), the bottom spectrum is plotted at the true values of delayed counts, while all other spectra are plotted with a vertical offset for visual clarity (see also Figs. S10–13).

(Fig. S14, Methods). We calculate a fit to all melting temperatures using the Simon-Glatzel formalism $T_m = T_{m0} \left( \frac{P_m - P_{m0}}{x} + 1 \right)^y$, where the melting points ($T_m$, $P_m$) are related to a reference melting point ($T_{m0}$, $P_{m0}$) with material-specific fit parameters $x$, $y$. With a fixed reference melting point of 1650 K at 0 GPa[51,52], we find best-fit values $x = 6.6 \pm 2.3$ and $y = 0.30 \pm 0.04$, resulting in a high-precision ($R^2 = 0.98$) melting curve that constrains a melting temperature of $4140 \pm 110$ K for Fe$_{0.94}$O at the core-mantle boundary pressure of 135.8 GPa (Fig. 6).

### Comparison with previous studies

The melting curve determined in this study agrees with melting temperatures reported by two previous experimental studies: low-pressure measurements[53] using the sinking of iron particles through a molten wüstite sample to determine melting from quenched sample analysis, and a 50 GPa measurement[11] using the disappearance of $B1$-Fe$_{0.99}$O reflections in x-ray diffraction measurements on the Fe-FeO

system. Agreement is also found with two previous studies that constructed melting curves using thermodynamic calculations of Gibbs free energies for solid and liquid states[5,12]. Our extrapolated melting curve further shows compatibility with reports of solid $B1$-Fe$_{0.96}$O up to 240 GPa and 4900 K[54]. One experimental study using changes in quenched sample texture reported considerably higher melting temperatures for Fe$_{0.94}$O with extreme (up to 1000 K) differences between the hottest solid and coolest liquid measurements[6].

In contrast, three previous experimental studies on similar compositions (Fe$_{0.94}$O to Fe$_{0.96}$O) reported lower melting temperatures (by ~350 K at 50 GPa), using visual observation of "fluid like motion" in sample surface texture[52,55], and emissivity−temperature discontinuities[7]. These techniques are sensitive to discontinuous changes in the sample's optical properties during phase transitions without structural information. We observe that the melting temperatures from these studies fall on a similar trendline as the order-disorder transition determined in this study (Fig. 6).

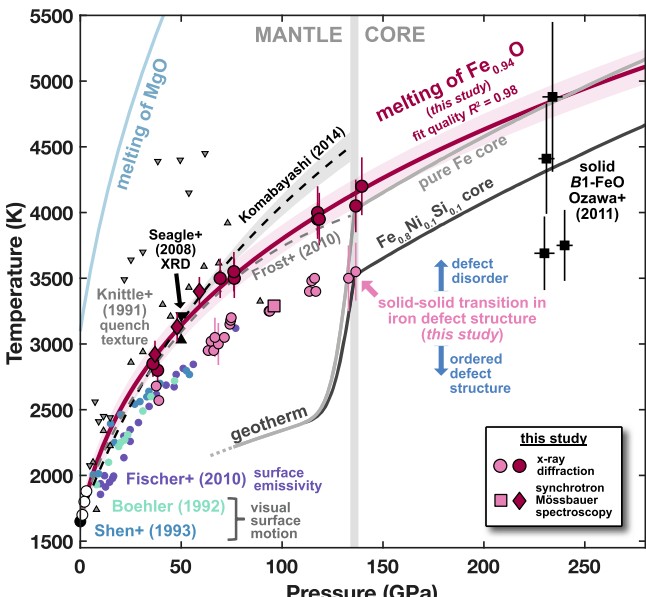

**Fig. 6 | Phase diagram of Fe$_{1-x}$O.** Crimson and pink points indicate melting and defect order transitions, respectively, measured in this study. Pink square represents the highest temperature reached in the highest-pressure SMS heating run where no signature of melting was observed. Melting points for Fe$_{1-x}$O from previous studies are shown with references indicated by labels, except for low-pressure[53] (open circles) and ambient pressure[51] (filled circle) studies. Core geotherms are constructed from melting studies using identical techniques[33,45,47]. Mantle geotherm and melting curve of MgO are taken from previous studies[62,79]. Source data are provided as a Source Data file.

We suggest the possibility that these studies detected the defect order-disorder transition, rather than a signature of melting. In our study, we generally observe plateaus in temperature versus laser power profiles at the order-disorder transition, suggesting discontinuous changes in properties like sample reflectivity or heat capacity[56]. Interestingly, one study[52] reported larger changes in surface texture "several hundred degrees" above the initial small changes at the reported melting temperatures, and further observed typical melt textures in quenched samples only when quenching from the higher transition temperatures. The authors may have detected both the order-disorder and the melting transitions but assigned the former as the latter. In contrast, reported data from a different study[7] suggest the samples may not have been heated to high enough temperatures to observe the melting transition above the order-disorder transition. Finally, a recent XRD study reported a melting curve similar to these three studies, though we note that the P-T conditions in that study used to study the liquid structure of FeO fall at or above our reported melting curve[57].

### Geophysical implications

The findings reported here affect multiple aspects of Earth's present-day lowermost mantle, as well as its early evolution. We first explore the viability of proposed solid FeO-rich ultralow velocity zones (ULVZs) at Earth's mantle base by comparing the updated FeO melting curve with CMB temperature estimates from previous work. Using core temperature profiles determined from recent melting studies of iron alloys using the same techniques as in this study[33,45,47], we see that the melting curve of iron predicts a CMB temperature of 4000 K[33], representing a moderate value across the range of recently reported iron melting curves[45]. The combined presence of 10 mol% each of nickel and silicon in the core, compatible with seismic and geochemical constraints[58-60], could lower the CMB temperature to 3500 K[45] (Fig. 6).

The melting temperature of FeO at the CMB (4140 K from this study) falls well above these estimates. Melting of FeO represents a lower bound on the solidus of magnesiowüstite phases proposed to exist in ULVZ assemblages[27], as addition of magnesium should raise the melting temperature of the (Mg,Fe)O solid solution[61,62]. The presence of silicates, on the other hand, is expected to produce at most only a mild decrease in the solidus temperature of such FeO-rich assemblages relative to the FeO end-member melting temperature[8], though the precise effect requires further study. This suggests that FeO-rich ULVZs, such as those containing iron-rich (Mg,Fe)O[24,26-28], could exist as solid structures in the present-day lowermost mantle. Given reports of high electrical conductivity and metallic-like behavior for solid FeO and iron-rich (Mg,Fe)O[32,63], solid FeO-rich ULVZs may exhibit higher bulk electrical and thermal conductivity than the surrounding mantle[14]. These proposed structures could provide a favorable mechanism for plume generation in the lowermost mantle and may also help explain observed variations in the length of day and nutations of Earth's rotation axis, both attributed to an electrically conductive layer in the lowermost mantle[64].

In addition, the discovery of a defect order-disorder transition in FeO up to lowermost mantle pressures may have broad implications for properties of FeO-rich regions at the CMB, as the structure of iron defects could strongly influence key physical properties like viscosity and conductivity[43,44]. We note that the temperature of the order-disorder transition at CMB pressure (~3550 K) coincides closely with recent temperature estimates for the region (Fig. 6). This finding reveals a novel mechanism for temperature variations at the mantle base to produce strong variations in physical properties, perhaps relating to observations of patchy or variably thick ULVZ regions[20,27]. These phenomena may be relevant even with the addition of magnesium into the (Mg,Fe)O solid solution, given reports of similar defect-produced satellite reflections in high-pressure, room-temperature XRD measurements of B1-(Mg$_{0.22}$Fe$_{0.78}$)O[65]. Finally, the melting curve of FeO can strongly influence models of magma ocean crystallization (e.g., refs. 8,9), which have relied on previous FeO melting temperature estimates[7] that were lower by ~350 K at 50 GPa and ~500 K at 136 GPa. The higher melting temperatures for FeO determined in this study thus imply a faster time scale for the crystallization of Earth's primordial magma ocean than previously suggested.

## Methods

### Materials

Sample material Fe$_{1-x}$O wüstite was synthesized using 95% enriched $^{57}$Fe in a gas-mixing furnace at ambient pressure[66]. The sample pellet was equilibrated at ~1575 K for two runs of 20 h each just above the iron-wüstite fugacity buffer (an oxygen potential of $10^{-9.6}$ atm)[67]. Sample composition and homogeneity were measured using a JEOL JXA-8200 electron microprobe. The ambient pressure lattice parameter was determined to be $a = 4.306(1)$ Å using x-ray diffraction[29]. The chemical composition of the material was computed using the lattice parameter and the relationship ($a = 3.856 + 0.478x$ for Fe$_x$O)[68], giving a composition of Fe$_{0.941(2)}$O. Samples in this study were taken from the same material batch used previously[29] to study the sound velocities of the material up to 94 GPa.

High pressure was achieved using diamond anvil cells (DACs) with diamond culet diameters ranging from 100 to 400 μm. Seats holding the anvils were composed of tungsten carbide on the upstream side of the DAC and cubic boron nitride on the downstream side in order to maximize the accessible 2θ scattering angle range for the XRD measurements. Rhenium gaskets serving as sample chambers were pre-indented to thicknesses of 30–50 μm from a starting thickness of 250 μm. Each gasket was drilled with an electron discharge machine to produce sample chamber diameters ranging from 35 to 165 μm, depending on the diamond culet size. Samples of starting thickness ~10 μm were sandwiched between flakes of dehydrated KCl with a

minimum thickness of 10 μm that served as both thermal insulation and a pressure transmitting medium. At least one ruby sphere was loaded inside the sample chamber without contact with the sample.

## Experimental methods

Once loaded, each DAC was heated in a vacuum oven for 24 h to minimize moisture in the sample chamber and subsequently sealed and compressed to its target pressure. Sample chamber pressures were estimated before and after each heating run from the fluorescence spectrum of the ruby spheres[69] and measured during the heating cycle for the XRD measurements. Sample heating locations were laser annealed at ~1500 K before the heating run to relax possible deviatoric stresses in the sample induced during synthesis and to determine sample coupling behavior with the infrared lasers. For XRD measurements, annealing time was ~1–2 min, while for SMS measurements, we annealed for ~5 min to allow for collection of a high-quality SMS time spectrum.

The melting of $Fe_{0.94}O$, hereafter referred to as FeO, is detected using a multi-technique approach that combines results from two in-situ methods: synchrotron x-ray diffraction (XRD) and synchrotron Mössbauer spectroscopy (SMS). XRD measurements are conducted at beamline 13-ID-D of the Advanced Photon Source (APS) using incident x-rays of energy 37 keV focused to a spot size of ~3 × 3 μm². We use double sided heating with infrared lasers (flat-top heating spot diameter ~10 μm[70] in a burst heating mode that collects alternating pairs of high-temperature ("hot") and 300 K ("quenched") measurements with exposure times between 1 and 4 s. SMS measurements are conducted at beamline 3-ID-B of the APS using incident x-rays prepared with a bandwidth of 1 meV at the 14.4125 keV energy of the nuclear resonance of $^{57}Fe$ (full-width half-maximum FWHM ~ 16 × 16 μm²)[71]. Double sided heating with infrared lasers (FWHM ~ 35 × 35 μm²) is controlled by a computer acquisition sequence that features an incremental ramp of laser power and collection of time-integrated Mössbauer signal intensity every 3 s. In-situ pressures are determined in the XRD heating runs from fits to integrated XRD patterns, while pressures before and after SMS heating runs are determined using ruby fluorescence at 3-ID-B (see below). The experimental techniques and measurement procedures are discussed previously in full detail[45].

## Calculating intensities from the x-ray diffraction patterns

We analyze a total of 1020 x-ray diffraction images in 19 separate heating runs over a pressure range of 30 to 140 GPa. XRD images are first azimuthally integrated using the software DIOPTAS[72]. We perform fits to the integrated patterns with the GSAS-II software package[73] for at least 6 measurements in each heating run in order to index the observed Bragg reflections and constrain unit-cell volumes of the two materials. We fit an additional ~160 measurements from four representative heating runs in order to assess sample pressure dependence on temperature during heating, referred to as thermal pressure (see below). All 1020 integrated patterns are further analyzed as follows. We analyze the evolution of sample and satellite reflection intensities during heating by integrating the area under all detectable reflections in each diffraction pattern, using a minimum of four sample and six satellite reflections. For the subsequent analysis, we consider the total sum of all satellite reflection intensities, as well as the total sum of sample reflection intensities normalized by that of the first measurement in the heating run.

In addition, we analyze the positions of satellite reflections, focusing on those surrounding the (2 0 0) reflection of the $B1$-FeO lattice. Following a previous high-pressure study[46], we identify these as (2 ± δ 0 0) reflections and calculate the values of δ for satellites in a given heating run using the $d$-spacing of each satellite reflection and the lattice parameter $a$, following the formula $\frac{1}{d^2} = \frac{h^2 + k^2 + l^2}{a^2}$ for a cubic lattice system. We do this for each heating run across the full pressure range of the study. This analysis shows that the satellite positions

appear to cluster around a set of particular δ values with no pressure dependence (Fig. 2). The scatter in δ values for each particular satellite position is comparable to the full-width half maximum of the satellite reflection peaks, represented by size of the shaded bars in Fig. 2. The consistent position of the satellites relative to the (2 0 0) $B1$ reflection across all pressures suggests that these satellites are caused by features of the defect-bearing $Fe_{0.94}O$ lattice, namely the long-range ordering of defects (superlattice), and not by a separate phase.

We further analyze background intensities of each diffraction pattern to identify melting by detecting liquid diffuse scattering signals, shown consistently to appear most strongly in low-angle regions around the low-order sample reflections[45,74]. We quantify background changes by selecting and integrating multiple background regions around the low-order sample reflections (Fig. 1, S1) where diffuse scattering is expected to be strongest ("low-angle region"), as well as a "high-angle region" (2θ ≅ 25°) where no diffuse scattering is expected (Fig. S1). This approach was shown previously[45] to produce the strongest sensitivity to liquid diffuse signals in $Fe_{0.8}Ni_{0.1}Si_{0.1}$. Background regions are chosen to be as far removed from Bragg reflections as possible. For subsequent analysis, we normalize low-angle background intensities by high-angle intensities for each hot and quenched pattern to analyze relative changes in the low-angle diffuse scattering region. This allows for independent analysis of hot and quenched diffuse signal intensities as they evolve during each heating run. The hottest sample temperature measured for each heating step is used in these analyses, with uncertainties estimated at ~150 K from the scatter around the melting curve fit to XRD and SMS results (Fig. 6). Reasonable transition temperature uncertainties are estimated from scatter in the data (Fig. 3).

## Synchrotron Mössbauer measurements

Synchrotron Mössbauer spectroscopy (SMS) is sensitive to the nuclear resonant signal produced exclusively by solid-bound $^{57}Fe$ atoms as their nucleus is excited by incident x-rays and subsequently decays from its first excitation state, characterized by a transition energy of 14.4 keV and an excitation lifetime of 141 ns. Signal intensity is related to the temperature-dependent Lamb-Mössbauer factor $f_{LM} = e^{-k^2 \langle u^2 \rangle}$, where $k$ is the wavenumber of the incident photon (7.30 × 10⁸ cm⁻¹) and $\langle u^2 \rangle$ is the mean-square displacement of the nucleus. As the sample is heated, a characteristic loss of signal intensity occurs when the mean-square displacement becomes very large within the excitation lifetime. This is the onset of melting[47].

We conduct four synchrotron Mössbauer heating runs at beamline 3-ID-B. The starting pressures at 300 K are determined from the ruby fluorescence spectrum[69], while the pressure increase at high temperature is determined from thermal pressures constrained by XRD measurements (see discussion below, Table S3). We begin each heating run by collecting a high-quality Mössbauer time spectrum while annealing the sample at ~1500 K for around 5 min[33,45]. These high-$P$-$T$ time spectra are collected in 24-bunch top-up mode, allowing for a timing window up to ~140 ns (Fig. S11). We further collect time spectra at ambient conditions on the starting sample material, as well as spectra at ~2.5 GPa and 300 K on samples measured at higher pressures in XRD heating runs at sector 13 (Figs. S12, 13). These low-$P$-$T$ spectra are collected in hybrid mode, allowing for a timing window up to ~600 ns. We fit all time spectra using the software CONUSS 2.3.0[75] to constrain the sample thickness at the start of the heating run. For the fits, we use estimates of the Lamb-Mössbauer factor based on previous measurements of $(Mg_{0.06}Fe_{0.94})O$[67] and the temperature dependence of the $f_{LM}$ determined for iron[76] (Tables S4–14). For the high-$P$-$T$ spectra, the best-fitting models require a magnetic site, consistently with ~25% relative weight, for all of the spectra. At 1500 – 1700 K, $B1$-FeO is not expected to exhibit magnetic ordering. Instead, we interpret this to be a signature of colder material further from the center of the laser heating spot that is within the radial tails of the x-ray

beam. As such, we renormalize the relative weights of the remaining three sites in the models to calculate the ferric iron concentration in the hot material. However, we note that the colder radial regions should contribute a signal to the time spectrum that is difficult to disentangle from the signal of the hot material, creating a challenge for quantitative interpretation of ferric iron content. Finally, from the best-fitting models to the time-domain Mössbauer spectra, we calculate and report energy-domain Mössbauer spectra, which visually show the width and intensities of the ferric and ferrous iron sites. In all fits, we explored thickness distributions and determined they are not required to improve the quality of the fits.

In the heating runs, we incrementally ramp up laser power on the sample over a series of 3-s intervals, while measuring the sample temperature using two different spectrometers[45,77] and total integrated intensities of the SMS signal within a particular time window. Temperature uncertainties for each 3-s collection are estimated from fluctuations in the high-frequency (~100 Hz) measurements from the FaSTeR spectrometer[33,77]. Fits are performed using the SIMX module in the software MINUTI 2.3.3[78], which models the Lamb-Mössbauer factor and thus signal intensity as a function of temperature given various experiment-specific parameters: the sample's effective thickness at the start of the heating run, constrained from fits to the Mössbauer time spectra, the time-window used for signal integration, and the sizes and shapes of the x-ray beam and laser hotspot[45]. Effective thickness is the dimensionless product of the numerical density of $^{57}$Fe atoms, the physical thickness of the sample, the nuclear resonant cross-section ($2.56 \times 10^{-22}$ m$^2$ for $^{57}$Fe), and the Lamb-Mössbauer factor. Influence of the sample's effective thickness at the start of the heating run is discussed in previous SMS melting studies[33,45,47]. The melting temperature is constrained by the fitting procedure, with uncertainties calculated as the root-mean-square of uncertainty from the fit and the average temperature uncertainty for each measurement. Changes in the sample chamber thickness, estimated from upstream and downstream ionization chambers monitoring total transmitted x-rays in situ[45], never exceed 2% before the onset of melting.

**Pressure determination and thermal pressure model**
For the XRD heating runs, we determine in-situ pressures of the defect order-disorder and melting transitions for all heating runs by first fitting diffraction patterns collected at the onset of each transition ($P1$, $P2$). From the refined unit-cell volumes of FeO and KCl, we calculate pressures of both materials using previously published thermal equations of state[48–50]. Transition pressures are calculated as the average of pressures given by the two materials, which consistently agree within ~2–3 GPa (Tables S1, S2), with uncertainty estimated from the difference in the pressures and uncertainty in the equations of state. Using pressure calculations from ~160 patterns across four heating runs, we calculate thermal pressures for FeO and KCl as the difference of each hot pressure both from the pressure of its corresponding quenched measurement and from the quenched pressure at the start of the heating run. We fit each of the resulting four thermal pressure data sets to determine a linear dependence of thermal pressure on temperature. We find good agreement between the two materials and the two calculation methods and note that larger scatter of FeO thermal pressures at high temperature are caused by the onset of the phase transitions (Fig. S14). We determine a thermal pressure model of $2.8 \pm 0.2$ GPa per 1000 K and apply it to predict the transition pressures in the SMS heating runs, using starting pressures from ruby fluorescence measurements[69] and a transition pressure uncertainty of 3 GPa.

## Data availability
All data used to construct Fig. 6 are available in the main text and supplementary materials, in the Source Data file, and in a Zenodo data depository (https://doi.org/10.5281/zenodo.10048208). Additional data are available from the corresponding author upon request. Source data are provided with this paper.

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

## Acknowledgements

We thank June Wicks for sample synthesis. We thank Paul Asimow, Michael Gurnis, and Zhongwen Zhan for valuable discussions. We are grateful to the National Science Foundation for supporting this work under EAR-1727020 and EAR-CSEDI—2009935 (J.M.J.). We acknowledge the JPL Strategic Research & Technology Development Program, "Venus Science Into The Next Decade". GeoSoilEnviroCARS and Sector 3 operations are partially supported by COMPRES (NSF-EAR-1661511). GeoSoilEnviroCARS is supported by the National Science Foundation—Earth Sciences (NSF-EAR-1634415) and Department of Energy- GeoSciences (DE-FG02-94ER14466). Use of APS is supported by the U.S. DOE, Office of Science (DE-AC02-06CH11357). SMS data collected during hybrid mode of the APS used a dual, fast-shutter spectrometer built by T.S.T. and supported by Laboratory Directed Research and Development (LDRD) funding from Argonne National Laboratory, provided by the Director, Office of Science, of the U.S. DOE under Contract No. DE-AC02-06CH11357.

## Author contributions

V.V.D. and J.M.J. designed the study. V.V.D. prepared the samples, led the XRD and SMS experiments, analyzed the data, produced the figures, and wrote the manuscript. J.M.J. and W.S. contributed to validating the data analysis. V.V.D., W.S., and D.Z. developed software for data analysis. J.M.J., D.Z., W.S., S.C., V.B.P., J.Z., T.S.T., and O.S.P. contributed to performing the experiments and editing the manuscript.

## Competing interests

The authors declare no competing interests.
