## [Peer Review File · Nature Communications]

REVIEWERS' COMMENTS

Reviewer #1 (Remarks to the Author):

The manuscript is very much improved on the original version and I think it is now suitable for publication in Nature Communications.

I would just make one point though regarding the authors responses – for their own information and even though this has no consequences for the paper.

The authors state,

“We further note that while the composition $\text{Fe}_{0.94}\text{O}$ implies a 6% concentration of iron vacancies, that does not necessarily imply a 12% concentration of ferric iron. Ferric iron can be located in either tetrahedral interstitial sites or in octahedral cation sites where it replaces a ferrous octahedral iron atom. The contribution to charge balancing is thus different for different ratios of interstitial to cation site concentrations of ferric iron.”

There have been arguments made like this in the past- from some strangely well respected scientists- that the cation deficiency in wüstite does not equate to the ferric iron content. But these proposals do not stand up to scrutiny. Firstly, it does not matter which site ferric iron or the vacancies enter, the oxygen charge balance has to be maintained. Any value of x in Fe_xO can only be satisfied by one ferric/ferrous ratio- regardless of where the ferric iron sits. If the authors don't believe this they should try and write down some particular site specific formulas and see if they can get different ferric/ferrous ratios to give them the same value of x . The site might influence coordination but it cannot influence how much Fe^{2+} and Fe^{3+} you need to arrive at a certain value of x . Secondly, if the authors really believe what they are writing here then they should reconsider how they report their stoichiometry. How did the authors determine their $\text{Fe}_{0.94}\text{O}$ stoichiometry? They got it from a published relationship between x and the unit cell parameters. But perhaps they should look up what this relationship is based on. They will find that the non-stoichiometry is determined through thermogravimetric measurements of – guess what- the ferric/ferrous ratio.

The authors also propose that

“Further, there have been suggestions that the iron vacancies themselves can hold charge.”

Certainly, many studies in electrical conductivity propose different types of charge carriers and a vacancy of course can be considered to be a local charge imbalance- but the charge is derived from the cations and anions and their oxidation states is all you need to calculate the charge neutrality.

Reviewer #2 (Remarks to the Author):

Review of Melting and defect transitions in FeO up to pressures of Earth's core-mantle boundary by Dobrosavljevic et al. submitted to Nature Communications.

This is a revised manuscript. In the earlier manuscript, the reviewers raised concerns with poor characterization of the satellite peaks which the authors claimed was caused by ordering of a superstructure in FeO. In the new revised manuscript, the authors made extensive additions to the results and discussions that support their interpretations. I would find this a reasonable revision and the authors were genuinely trying to address the reviewers' concerns. What I particularly appreciate is that the authors have added new data showing the satellite peak positions are linked to the main peaks of FeO at any pressure, demonstrating that the satellite peaks are from the FeO structure, although the detailed peak assignments are very challenging due to a number of possible factors.

Previous experimental work (ref. 7) reported lower melting temperatures of Fe_{0.40}O than this study under high pressures. The order-disorder transition temperatures in this new experiment coincide with the previous reports of melting points. This gives a reasonable explanation for the existing disagreements. Although the detailed analysis of the previous works is not possible by the authors, the authors have provided firm evidence of the presence of the order-disorder transition in Fe_{0.40}O which is identical to the sample composition in the previous work, and therefore the argument that ref. 7 should have observed it sounds reasonable. In conclusion, the authors claim that FeO melting points at high pressures should be greater than previously thought sounds robust.

The determination of the melting points of FeO at the CMB pressure will have high impacts on the Earth and Planetary Sciences community. FeO is a fundamental component of the mantle and therefore, any petrological discussion at the CMB will inevitably rely on the result of this paper. The examples that are listed in the discussion section are well summarized. I believe that this paper if published will be cited many times over a long period of time.

The only comment/suggestion from me is: can the authors argue that the recent paper by Morard et al. (2022) could also have observed the order-disorder transition as they observed diffuse scatterings from the sample?

Reviewer
Author replies

Reviewer 1

Reviewer #1 (Remarks to the Author):

The manuscript is very much improved on the original version and I think it is now suitable for publication in Nature Communications.

We thank the reviewer for the positive feedback regarding the revised manuscript and their support for its publication in this journal.

I would just make one point though regarding the authors responses – for their own information and even though this has no consequences for the paper.

The authors state,

“We further note that while the composition $\text{Fe}_{0.94}\text{O}$ implies a 6% concentration of iron vacancies, that does not necessarily imply a 12% concentration of ferric iron. Ferric iron can be located in either tetrahedral interstitial sites or in octahedral cation sites where it replaces a ferrous octahedral iron atom. The contribution to charge balancing is thus different for different ratios of interstitial to cation site concentrations of ferric iron.”

There have been arguments made like this in the past- from some strangely well respected scientists- that the cation deficiency in wüstite does not equate to the ferric iron content. But these proposals do not stand up to scrutiny. Firstly, it does not matter which site ferric iron or the vacancies enter, the oxygen charge balance has to be maintained. Any value of x in Fe_xO can only be satisfied by one ferric/ferrous ratio- regardless of where the ferric iron sits. If the authors don't believe this they should try and write down some particular site specific formulas and see if they can get different ferric/ferrous ratios to give them the same value of x . The site might influence coordination but it cannot influence how much Fe^{2+} and Fe^{3+} you need to arrive at a certain value of x . Secondly, if the authors really believe what they are writing here then they should reconsider how they report their stoichiometry. How did the authors determine their $\text{Fe}_{0.94}\text{O}$ stoichiometry? They got it from a published relationship between x and the unit cell parameters. But perhaps they should look up what this relationship is based on. They will find that the non-stoichiometry is determined through thermogravimetric measurements of – guess what- the ferric/ferrous ratio.

The authors also propose that

“Further, there have been suggestions that the iron vacancies themselves can hold charge.”

Certainly, many studies in electrical conductivity propose different types of charge carriers and a vacancy of course can be considered to be a local charge imbalance- but the charge is derived

from the cations and anions and their oxidations states is all you need to calculate the charge neutrality.

We thank the reviewer for providing discussion on the details of the ferric iron and vacancy concentration relationship. We appreciate the thoughtful and detailed explanation provided. Upon re-evaluation of site-specific formulas as suggested by the reviewer, we agree with the conclusion that indeed a specific value of iron deficiency in the Fe_{1-x}O lattice (x) should in principle correspond to one specific ratio of ferric to ferrous iron. For $x = 6\%$, as in $\text{Fe}_{0.94}\text{O}$, this should be charge balanced by a ratio of 12 ferric to 82 ferrous iron atoms, or $\sim 12.8\%$, in close agreement with the ferric iron content estimated for our starting material from the SMS time spectra. Beyond that statement in our manuscript, we do not further comment on specific implied vacancy concentrations, and we appreciate the reviewer stating that the present discussion is for our information and does not require modification to the manuscript. At the same time, we would like to suggest that as a defect-bearing material exhibiting strong electron correlations, the traditional notions of individual “ferrous” and “ferric” iron atoms may not perfectly capture the distributions of charges within the lattice. With this consideration, we generally use the term “ferric-like” within the manuscript in our description of SMS results. Finally, we would like to point out that the study by McCammon and Liu (1984) on the relationship between the lattice parameter and non-stoichiometry compiles a series of earlier studies, some of which additionally include wet chemistry to determine the total content of iron, though the reviewer’s point stands that the results from many studies contributing to the compilation are indeed based on thermogravimetry.

Reviewer 2

Reviewer #2 (Remarks to the Author):

Review of Melting and defect transitions in FeO up to pressures of Earth's core-mantle boundary by Dobrosavljevic et al. submitted to Nature Communications.

This is a revised manuscript. In the earlier manuscript, the reviewers raised concerns with poor characterization of the satellite peaks which the authors claimed was caused by ordering of a superstructure in Fe_xO . In the new revised manuscript, the authors made extensive additions to the results and discussions that support their interpretations. I would find this a reasonable revision and the authors were genuinely trying to address the reviewers’ concerns. What I particularly appreciate is that the authors have added new data showing the satellite peak positions are linked to the main peaks of FeO at any pressure, demonstrating that the satellite peaks are from the FeO structure, although the detailed peak assignments are very challenging due to a number of possible factors.

Previous experimental work (ref. 7) reported lower melting temperatures of $\text{Fe}_{0.4}\text{O}$ than this study under high pressures. The order-disorder transition temperatures in this new experiment coincide with the previous reports of melting points. This gives a reasonable explanation for the existing disagreements. Although the detailed analysis of the previous works is not possible by

the authors, the authors have provided firm evidence of the presence of the order-disorder transition in Fe_{0.40}O which is identical to the sample composition in the previous work, and therefore the argument that ref. 7 should have observed it sounds reasonable. In conclusion, the authors claim that FeO melting points at high pressures should be greater than previously thought sounds robust.

The determination of the melting points of FeO at the CMB pressure will have high impacts on the Earth and Planetary Sciences community. FeO is a fundamental component of the mantle and therefore, any petrological discussion at the CMB will inevitably rely on the result of this paper. The examples that are listed in the discussion section are well summarized. I believe that this paper if published will be cited many times over a long period of time.

We are sincerely thankful to Reviewer 2 for the positive feedback regarding the revisions to the manuscript and for their support of its publication in this journal. We appreciate the specific positive feedback on the additional analysis regarding satellite peak positions as an important piece of evidence supporting the interpretation of the data. To strengthen the manuscript further, we have moved the relevant figure (previously Fig. S2) into the main text as the new Fig. 2.

The only comment/suggestion from me is: can the authors argue that the recent paper by Morard et al. (2022) could also have observed the order-disorder transition as they observed diffuse scatterings from the sample?

It is difficult to interpret the results of Morard et al. (2022) in the context of the order-disorder and melting transitions, due to the limited amount of XRD data reported in that study, which includes only a small number of integrated diffraction patterns from one heating run shown in a figure inset.